# Fruit Peels as a Sustainable Waste for the Biosorption of Heavy Metals in Wastewater: A Review

**DOI:** 10.3390/molecules27072124

**Published:** 2022-03-25

**Authors:** Dora Luz Gómez-Aguilar, Juan Pablo Rodríguez-Miranda, Octavio José Salcedo-Parra

**Affiliations:** 1Departamento de Química, Universidad Pedagógica Nacional, Bogotá 110231, Colombia; 2Facultad del Medio Ambiente y Recursos Naturales, Universidad Distrital Francisco José de Caldas, Bogotá 110231, Colombia; jprodriguezm@udistrital.edu.co; 3Facultad de Ingeniería, Universidad Distrital Francisco José de Caldas, Universidad Nacional de Colombia, Bogotá 110231, Colombia; ojsalcedop@unal.edu.co or

**Keywords:** industrial wastewater, biosorption, fruit peels, heavy metals

## Abstract

One of the environmental challenges that is currently negatively affecting the ecosystem is the continuous discharge of untreated industrial waste into both water sources and soils. For this reason, one of the objectives of this qualitative study of exploratory-descriptive scope was the review of scientific articles in different databases—Scopus, Web of Science, and Science Direct—published from 2010 to 2021 on the use of fruit peels as a sustainable waste in the removal of heavy metals present in industrial wastewater. For the selection of articles, the authors used the PRISMA guide as a basis, with which 210 publications were found and 93 were compiled. Considering the reported work, a content analysis was carried out using NVivo 12 Plus and VOSviewer 1.6.17 software. The results show that the fruits mentioned in these publications are lemon, banana, mango, tree tomato, pineapple, passion fruit, orange, coconut, avocado, apple, lulo, and tangerine. However, no studies were found with lulo and tree tomato peels. On the other hand, the heavy metals removed with the selected fruit peels were Pb^+2^, Cr^+3^, Cr^+6^, Ni^+2^, Cd^+2^, As^+5^, Cu^+2^, and Zn^+2^.

## 1. Introduction

Rapid population growth has led to an increase in industrial activities that have generated different types of wastewater that must be treated before being discharged into water sources in order to avoid a negative impact on the ecosystem due to the different substances of organic and inorganic nature and microorganisms present in this type of water [1].

In relation to the above, the World Water Development Report 2017 states that more than 80% of wastewater is discharged without any treatment, causing dead areas without oxygen in the seas and oceans, affecting the fishing industry sector and the food cycle derived from it [2].

The United Nations World Water Development Report 2020 [3] specifies that the detriment of water sources puts at risk the fulfillment of Sustainable Development Goal (SDG) number 6 of the 2030 Agenda, whose purpose is to provide basic sanitation for the entire population in the next 10 years, given that, at present, 4.2 billion people lack access to these safe systems.

On the other hand, in Latin America, 70% of wastewater is discharged without any treatment into rivers, generating problems for the environment and public health. Likewise, 80% of the inhabitants live in cities, and the remaining 20% are located near contaminated surface water sources [4].

This statement is corroborated by a review carried out for heavy metals in 40 estuaries in Latin America located between the Atlantic and Pacific coasts where six metals were found in sediments and in some surface water samples [5]: Ni, Hg, Cu, Pb, Cr, Cd. The six referenced metals, with the exception of Ni, (Hg, Cu, Pb, Cr, and Cd) are considered by the United States Environmental Protection Agency (USEPA) as the most relevant in terms of their impact on public health [6]. In Colombia, there is the presence of mercury in the sediments of the Ciénaga Grande de Santa Marta, with concentrations of 129 mg Hg Kg^−1^, and in the tissues of some fish species such as Eugerres plumieri and Mugil incilis [5]. Regarding the concentration of Hg in sediments, it is observed that it exceeds the maximum limits established for this metal by the international regulations of Canada, Holland, and the United States, which are: 0.13 mg Hg Kg^−1^, 0.3 mg Hg Kg^−1^, and 0.7 mg Hg Kg^−1^, respectively. It is important to highlight that the Hg concentration is above the 0.7 mg Hg Kg^−1^ probable effect level (PEL) of the United States and “may have probable effects on biota”. Therefore, the sediments of the Ciénaga Grande de Santa Marta could represent a risk to the ecosystem, including the population [7]. 

Additionally, the epicenter located 137 km away from the Ciénaga is the Bay of Cartagena, which is considered by a group of scientists in Colombia as a “terminal patient” due to the high concentration of heavy metals (Hg, Cr, Cr, Cd, Ni, and Pb), garbage, and fecal coliforms, causing various diseases in the digestive system, dermatological lesions, and altered metabolic profiles in the population [8]. It is also worth mentioning that an analysis of mercury in the blood of the inhabitants of Baru found an alert level for the presence of this metal. Among some of the pathologies associated with the presence of heavy metals in humans are renal failure, liver damage, hepatitis, heart failure, asthma, birth defects, respiratory failure, mental disorders, hypertension, and cancer [9]. 

Mining, textile, electroplating, metallurgy, foundry, alloys, steel, metallic corrosion, paints, batteries, electronics, tanneries, agriculture, livestock activities, landfills, and energy production are among the industries that generate heavy metals discharges [6,10]. According to a report by the Superintendence of Domiciliary Public Services, it is observed that there was an increase in the percentage from 25.0% to 42.8% in wastewater treatment from 2010 to 2018. Consequently, Colombia is ranked fourth among Latin American countries, below Chile (99.9%), Mexico (57%), and Brazil (43%) [11]. In Colombia, the most prevalent wastewater treatment is secondary, in which stabilization ponds are used at 44%, extended aeration systems at 9.4%, and biological filters at 7% [12]. 

On the other hand, heavy metals as inorganic pollutants mostly present in wastewater require technologies such as conventional, advanced, and non-conventional methods. For the first, there are adsorption methods (activated carbon and carbon nanotubes) and chemical precipitation using alkaline solutions; for the second, there are ion exchange and membrane filtration methods (microfiltration, nanofiltration, ultrafiltration, reverse osmosis, and electrodialysis), among others. The advantage of conventional and advanced methods is their high efficiency, but among the disadvantages are their high cost of implementation and the high volume of sludge that can be generated, especially in the chemical precipitation method [10].

Non-conventional methods for the removal of heavy metals include bioremediation, phytoremediation, hydrogels, and fly ash. For living and non-living organisms, biomass is used, which allows the sorption of various types of contaminants, such as those mentioned above. Additionally, a series of investigations have been reported in the literature which have sought to reevaluate lignocellulosic materials or agricultural residues as non-living biomasses (shells, bark, stems, leaves, seeds, etc.); likewise, the use of by-products from the tannery sector, such as hair, shavings, among others, has been reported as a technology used in sorption phenomena [13].

This article socializes the findings of scientific studies published from 2010 to 2021 on the fruit peels used in Colombia (introduced and native) as a sustainable waste for the removal of heavy metals, which represents an opportunity to be used in industries. The analysis was made because there have not yet been published reports that condense the research on this type of material in order to specify the optimal adsorption conditions, the percentage of inedible biomass based on the fruit, efficiency percentages against different heavy metals in synthetic and/or real wastewater matrices, selectivity of the peels in relation to the mitigation of metals, lignocellulosic composition of the selected fruit peels, and the comparison between different fruit peels based on their efficiency for the mitigation of heavy metals in real water.

Finally, it is proposed that in industries worldwide, particularly in Colombia, companies implement sorption technology with fruit peels in their wastewater so that they generate a reuse of the same complying with the requirements demanded in Resolution 1207 of 2014 by the Ministry of Environment and Sustainable Development (MESD), which states that this type of water could be used for agricultural or industrial use [14]. This will allow our country and others to not be lagging behind in terms of this application, given that, at this moment, Colombia appears to have 0% with respect to wastewater reuse, compared with Israel (87%), Mexico (60%), South Africa (24%), Peru (4%), and Chile (0.7%) [15]. The advantage of reusing this type of water is that it allows dealing with its scarcity in the medium term due to the decrease in pressure on the extraction of surface and groundwater sources. This also allows countries to acquire a circular economy.

## 2. Materials and Methods

### 2.1. Documentary Analysis

The study is qualitative and exploratory-descriptive in scope [16]. A review of scientific articles from 2010 to 2021 was conducted in the databases of Scopus, Web of Science, and Science Direct. A list of keywords or thesauri in English and Spanish languages (e.g., heavy metals, bioadsorption, removal, wastewater, and fruit peels) was configured. Additionally, the Preferred Reporting Items for Systematic Reviews and Meta-Analyses (PRISMA) [17] guide was used as a basis to select articles related to the topic under investigation, duplication of documents, and probable gray literature.

### 2.2. Content Analysis

With respect to the research products found and selected, the titles, keywords, abstracts, and content were read, choosing those related to fruit peels as sorbents for treating industrial wastewater contaminated with heavy metals. Once selected, NVivo 12 Plus (trial version) and VOSviewer 1.6 software (free to use) were used to perform the analysis. The content analysis was carried out using the free 17 software, focusing on the co-occurrence of terms, journals, countries, paper specifications (author, country, institution/organization), years of publication, and optimal sorption conditions. Therefore, parameters such as absorbent dose, temperature, particle size, and speed of agitation were considered, as well as gaps and future research in this perspective. 

## 3. Results and Discussion

### 3.1. Selected Fruit Peels

First, several documents related to nutritional information on the most consumed foods by the national population were reviewed. A study conducted by the National Survey of Nutritional Status ENSIN [18] on the feeding practices of people with an interest in nutrition and public health in the population between 5 and 64 years of age shows that the first group of foods is the consumption of fruits, 66.8%, followed by dairy and meats, with percentages of 61.0% and 57.3%, respectively (see Figure 1).

With the selected product, we proceeded to identify those fruits with the highest domestic consumption at the national level. In Colombia (2007), 433 species of edible fruits were reported, positioning the country as the first in the world in biodiversity of these foods per square kilometer. This report was provided by the Colombian Corporation for Agricultural Research (CORPOICA), currently known as AGROSAVIA, since 1994 [19]. 

There are 54 fruits of high consumption in Colombia: lemon (*Citrus limón*), banana (*Musa paradisiaca*), mango (*Mangifera indica*), guava (*Psidium guajava*), tree tomato (*Solanum betaceum*), blackberry (*Rubus ulmifolius*), pineapple (*Ananas comosus*), passion fruit (*Passiflora edulis*), orange (*Citrus sinensis*), coconut (*Cocos nucífera*), avocado (*Persea americana*), papaya (*Carica papaya*), apple (*Malus pumila*), lulo (*Solanum quitoense*), tangerine (*Citrus reticulata*), soursop (*Anona muricata*), strawberry (*Fragaria spp*), curuba (*Passiflora tarminiana*), melon (*Cucumis melo*), watermelon (*Citrullus lanatus*), grape (*Vitis vinifera*), passion fruit (*Passiflora ligularis*), borojo (*Alibertia patinoi*), pear (*Pyrus communis*), plum (*Prunus doméstica*), papayuela (*Vasconcellea pubescens*), loquat (*Achras sapote*), peach (*Prunus pérsica*), mamoncillo (*Melicoccus bijugatus*), sapote (*Pouteria sapota*), uchuva (*Physalis peruviana*), chontaduro (*Bactris gasipaes*), bananito (*Musa AA Si- monds*), feijoa (*Acca sellowiana*), cherry (*Prunus subg. Cerasus*), carambolo (*Averrhoa carambola*), kiwi (*Actinidia deliciosa*), guama (*Inga edulis*), grapefruit (*Citrus aurantium*), badea (*Passiflora quadrangularis*), arazá (*Azara*), noni (*Morinda citrifolia*), fig (*Ficus carica*), cashew (*Anacardium occidentale*), mamey (*Pouteria sapota*), lime (*Citrus aurantiifolia*), custard apple (*Annona cherimola*), pomarrosa (*Syzygium jambos*), pitahaya (*Selenicereus undatus*), mangosteen (*Garcinia mangostana*), copoazú (*Theobroma grandiflorum*), anón (*Annona squamosa*), breadfruit (*Artocarpus altilis*), and currant (*Ribes rubrum*) [20]. From the report of this number of fruits, a selection was made regarding those with the highest domestic consumption at the national level, obtaining 15 fruits, which are the following: lemon, banana, mango, guava, tree tomato, blackberry, pineapple, passion fruit, orange, coconut, avocado, papaya, apple, lulo, and tangerine [20]. Subsequently, they were classified according to the inedible shell-to-seed ratio [21] (see Figure 2).

It should be noted that 12 out of the 15 fruits referenced in Figure 2 were selected since papaya, guava, and blackberry are those with the lowest inedible part value and also because the proportion of peel in these fruits is considered lower than that of seeds. In Colombia, there is a research gap that leads to determining the percentage of inedible parts separately between seed and peel per fruit based on its variety.

Therefore, fruits such as passion fruit, coconut, lemon, mango, pineapple, lulo, orange, tree tomato, avocado, banana, tangerine, and apple were selected for review in the mentioned databases since these are the ones that present a higher shell biomass in relation to the seeds. This is summarized in Figure 3.

From the three databases (Scopus, Web of Science, and Science Direct), the results presented in Figure 4 were obtained by considering the Preferred Reporting Items for Systematic Reviews and Meta-Analyses (PRISMA) guidelines.

With respect to the compilation obtained in the three databases, Table 1 and Figure 5 were designed to show which of the 12 fruit peels are related, with the objective of identifying which of these have been the most researched and published, as well as those that have no publication and represent a research gap.

Table 1 and Figure 5 show that orange peels, followed by banana and coconut peels, are the ones that presented the highest number of publications, between 25.8%, 21.5%, and 15.05%, respectively, while lulo and tree tomato peels had no publications.

On the other hand, in the co-occurrence of the terms “heavy metals”, “wastewater”, “fruit peels”, and “adsorption”, using the metadata of Scopus/VOSviewer 1.6.17, Figure 6 was obtained, in which we can visualize the nodes representing the most used words by the authors. Thus, coconut, banana, and orange prevail from highest to lowest; also, some relationships between them can be visualized (agricultural waste, batch scale research with synthetic and/or real wastewater matrices). According to the color scale, publication predominance occurs from 2014 onwards.

### 3.2. Generalities of the Articles Found and Selected

Table 2 illustrates some generalities according to the origin and the type of major perennial fruits and transitory fruits introduced in Colombia with respect to the 12 fruits selected and described in numeral 3.1.

Table 2 shows that, of the 12 fruits selected, 11 have been introduced in Colombia, with the exception of lulo. This is a species of the Solanaceae family that is widely distributed in the Andes mountain range and that grows spontaneously in the undergrowth near streams. Its cultivation is prevalent in Peru, Ecuador, Colombia, Panama, Costa Rica, and Honduras. In addition, lulo is considered a promising crop due to its nutraceutical value, which makes it desirable in national and international markets [114]. 

Fruits such as lemon, mango, orange, coconut, avocado, and tangerine are considered major perennial fruits, i.e., they are permanent species that are economically and socially important. On the other hand, bananas, tree tomatoes, pineapples, passion fruit and lulo, are considered transient fruits, since the vegetative and productive cycle of these species does not exceed three years [19].

Regarding the countries where the terms “heavy metals” and “wastewater” and “fruit peels”, and “adsorption” have been published and mentioned, the co-occurrence of words is observed using Scopus/VOSviewer 1.6.17 metadata in Figure 7.

In Figure 7, it is observed that India is the country with the highest node, indicating that it has the highest number of publications related to the removal of heavy metals in wastewater using fruit peels from 2015 onwards. In turn, it is appreciated that there is a co-occurrence between Ghana and the Netherlands. 

In the same way, co-occurrence between authors, organizations, and journals was specified, using for this purpose the search equation described above with Scopus/VOSviewer 1.6.17 metadata, obtaining Figure 8, Figure 9 and Figure 10, respectively.

Figure 8, Figure 9 and Figure 10, reference authors, organizations, and journals, respectively, and show the predominance of researchers such as Lens P.N.L. and Shiewer S; likewise, organizations such as the Department of Civil Engineering and the Department of Civil and Environmental Engineering are highlighted. Regarding the most mentioned journals, they are the *Journal of Hazardous Materials* and *Bioresource Technology*, according to the visualization of the nodes that stand out in the graphic. There are no relationships between authors or organizations among themselves, and in the journals, it is deduced that they have been published since 2012.

### 3.3. Colombian Solid Waste

In Bogotá, D.C., 0.87 kg/inhabit/day of solid waste is being generated; this value is within the range of 0.19 to 1.81 reported by Latin American countries [116]. It is important to clarify that 61% of this solid waste is generated by organic waste, as shown in Figure 11 [117].

Based on the analysis carried out, currently, 35.21% of the solid waste generated in Bogotá, D.C., is used, and 64.78% is deposited in sanitary landfills (see Figure 12). Based on a study conducted for Colombia on alternative techniques for the treatment, final disposal, or use of solid waste, these are being used for energy generation, recycling, mechanical treatment, landfilling, and composting. It should be noted that in other countries, waste is still discharged or disposed of by incineration or used as raw material for the extraction of active substances and animal feed, which leads to serious environmental contamination and a great waste of resources [118]. In Colombia, 83% of the solid household waste generated is discharged into sanitary landfills. For this reason, their use should be improved, given that in 2030 there could be sanitary emergencies and a high emission of greenhouse gases (GHG) such as methane, carbon dioxide, water vapor, and hydrogen [119].

It is also important to highlight that in order to avoid the abovementioned, countries could opt for some of the programs formulated at the Earth Summit (an event held in 1992 by the UN in Rio de Janeiro) and established for waste management: minimizing waste and maximizing its reuse and ecological recycling. These actions encourage sustainable development and a rational ecology for the 21st century [120].

This article proposes the use of a solid organic waste such as fruit peels, which is a lignocellulosic bioadsorbent composed of cellulose polymers, hemicellulose, and lignin for the removal of heavy metals [121]. Among the fruits selected in the documentary analysis from 2010 to 2021 are those referenced in numeral 3.1: passion fruit, coconut, lemon, mango, pineapple, lulo, orange, tree tomato, avocado, banana, tangerine, and apple. From this list of fruits, four are considered citrus, according to statistical data from the Food and Agriculture Organization of the United Nations (FAOSTAT), which states that world citrus production in 2010 exceeded 100 million tons [122]. It is also relevant to point out that this type of citrus waste can account for up to 50% of the weight of the fruit, as shown in Figure 2.

With respect to the content analysis, regarding the selected fruits, Figure 13 was obtained using scientific articles from Scopus, Web of Science, and Science Direct metadata collections from 2010 to 2021, in which the names of the authors, year of publication, institutions, countries, and journals that have published on the removal of heavy metals with fruit peels can be observed. Besides, some optimal adsorption conditions (pH, pHpzc, temperature, agitation, physical and/or chemical modifications, particle size), type of wastewater matrices, percentage of contaminant removal, initial concentration, desorption, bioadsorbent reuse cycles, selectivity, and explanation of adsorption processes (isotherms, kinetics, maximum adsorption capacity, and adsorption mechanisms) are specified in the nodes.

Subsequently, the qualitative analysis was performed with NVivo 12 plus software, with which the word cloud-adsorption-removal-heavy metals-wastewater was obtained, (see Figure 14). In this figure, some words such as adsorption, biosorption, removal, synthetic wastewater, heavy metals, orange, banana, coconut, chromium, cadmium, kinetics, isotherms, desorption, and concentration stand out. As a result, the authors used these words for the documentary analysis of each fruit peel in the mentioned databases.

Based on the information obtained in Figure 13 and Figure 14, the authors elaborated Figure 15, Figure 16, Figure 17, Figure 18 and Figure 19 in which the optimal adsorption parameters for each fruit peel selected in numeral 3.1 are established. Regarding the heavy metals removed from the synthetic wastewater matrices, these figures illustrate a range of particle size, adsorbent dosage, physical or chemical modification of the adsorbent, optimum pH, optimum temperature, contact times, desorption with the different extractive solutions used, and efficiency percentages.

Table 3 and Table 4 show the chemical characterization (organic and inorganic) of orange peel in terms of elemental composition (carbon, hydrogen, nitrogen, and sulfur); lignocellulosic composition (cellulose, lignin, pectin, and hemicellulose), highlighting that this bioadsorbent has 97.83% of organic matter and the remaining 2.17% of inorganic matter, where acid and basic oxides predominate.

Table 5 reports the functional groups present in orange peel [123], with the objective of determining the groups responsible for the interaction with heavy metals for their respective removal.

The efficiency reported in Figure 15, Figure 16, Figure 17, Figure 18 and Figure 19 of the selected fruit peels ranges from 75% to 100% due to the abundance of hydroxyl and carboxyl groups present in the composition of lignin, cellulose, hemicellulose, pectin, chlorophyll pigments, and low molecular weight hydrocarbons [118]. Some authors specify that the functional groups responsible for these interactions are carboxyl groups [125,126,127].

It is important to note that this type of non-conventional technology of bioadsorption with fruit peels is proposed as a technique that could be used by industries that discharge heavy metals in their wastewater (see Table 6).

### 3.4. Biorefinery of Fruit Peels

Fruit peels are being used as a source of bioactive compounds of interest because they have a greater potential to be reused in human food or in the pharmaceutical industry due to the presence of compounds such as carotenes, total polyphenols, flavonoids, and anthocyanins, which are characterized by their antioxidant activity. Among the products obtained are essential oils, edible oils, natural pigments, food additives, anticancer compounds, enzymes, bioethanol, and production of biodegradable plastic, among others [118,129] (see Figure 20).

## 4. Technoeconomic Aspects and Future Research

The information shown in Figure 15, Figure 16, Figure 17, Figure 18 and Figure 19 is relevant in order to be able to reproduce in the first instance, with pilot models, these experimental conditions with real wastewater matrices (see Table 6), industrial activities that generate heavy metals, which will allow us to know the possible interferences that may occur, and thus to estimate again efficiency and desorption parameters, which in the future could be replicated on an industrial scale, which requires “significant financial and technological effort to be able to really carry it out” [118].

In the literature, there are only reports of heavy metals removal with industrial wastewater from the electroplating and battery industries and tanneries in pilot-scale designs. Metals such as copper [130], molybdenum [131], and chrome (VI) [132] were removed from wastewater from electroplating plants [132] in the electroplating wastewater, while copper, cadmium, and lead were mitigated in the battery industry [100], and finally chrome (VI) [133] was removed in the wastewater from the tanning industry [133]. Therefore, further research is required with these lignocellulosic materials with real wastewaters other than those mentioned above, such as mining, metallurgy, glass, chemicals, wood protection, plastics, paper, rubber, alloys, and electronics.

In this section, it is also important to highlight the desorption process of the metals with respect to the bioadsorbent. Given that, as described in Figure 15, Figure 16, Figure 17, Figure 18 and Figure 19, the extracting solutions that have been used are HCl, HNO_3_, H_2_SO_4_, NaOH, and EDTA. This makes it possible to separate the metal ions from the bioadsorbent so that this lignocellulosic material is again incorporated into the removal cycle of the metals present in the wastewater, which has the advantage of making the removal process cheaper in industries. Therefore, research should specify the maximum number of reuse cycles of the bioadsorbent in order to obtain a high percentage of efficiency.

In addition, this will contribute to reducing organic solid waste in sanitary landfills, thus avoiding more GHG generation and resulting in less impact on global warming. At the same time, the use of this type of non-conventional technology on an industrial scale for industrial wastewater treatment has become an opportunity that will allow companies to implement sustainable development by improving their social, economic, and especially environmental aspects (see Figure 21).

The above is corroborated by the ideas outlined in the Figure 22 word tree, which was obtained from NVivo 12 plus software (https://www.qsrinternational.com/nvivo-qualitative-data-analysis-software/support-services/nvivo-downloads), accessed on 18 January 2021.

Figure 22 shows that it is a challenge to prepare new adsorbent materials that are better in terms of being efficient and environmentally friendly—in this case, the use of fruit peels. The aforementioned takes advantage of the properties of these lignocellulosic materials, as well as their low cost and zero toxicity [118].

On the other hand, the importance of reusing wastewater that is free of heavy metals in this type of water is highlighted, as shown in Figure 21, which will contribute to meeting target 6.3 of the Sustainable Development Goal (SDG 6) for water and sanitation of the 2030 Agenda.

## 5. Conclusions

The documentary analysis from 2010 to 2021 in the different databases mentioned in the document indicates that of the 12 selected fruits, orange peels, followed by bananas and coconut peels, are the peels with the highest number of publications, with 25.8%, 21.5%, and 15.05%, respectively, in relation to the removal of heavy metals in industrial wastewater, while lulo and tree tomato peels are not in the reported work. Considering the above information, it is inferred that it is necessary to develop research using these two species. The lulo is considered a native species of Colombia and is characterized by its high nutraceutical value, while the tree tomato is an introduced species from Peru. Both are classified as transitory species.

The use of this non-conventional technology—the bioadsorption of heavy metals with fruit peels for the treatment of industrial wastewater on an industrial scale—represents an opportunity that will allow companies to implement sustainable development by improving their social, economic, and environmental aspects. As proposed in the challenges at the research level, the shells are constituted as a natural, effective, and environmentally friendly adsorbent.

Considering the documentary analysis of fruit peels as a sustainable waste for the biosorption of heavy metals in wastewater from 2020 to 2021, it is established that, to date, seven metals have been removed (Pb, Cr, Ni, Cd, As, Cu, Zn) out of the thirteen that, according to the World Health Organization, have been specified as those with the greatest impact on health and the environment. This makes it possible to infer that it is necessary to continue research on the removal of the remaining heavy metals (Hg, Mn, Co, Ti, Fe, Sn) in industrial wastewater using biomasses with the fruit peels selected in this article.

## Figures and Tables

**Figure 1 molecules-27-02124-f001:**
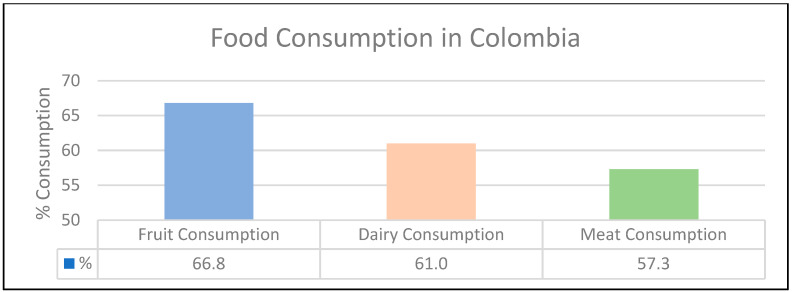
Food Consumption in Colombia. Reprinted with permission from ref. [18]. 2010 ICBF.

**Figure 2 molecules-27-02124-f002:**
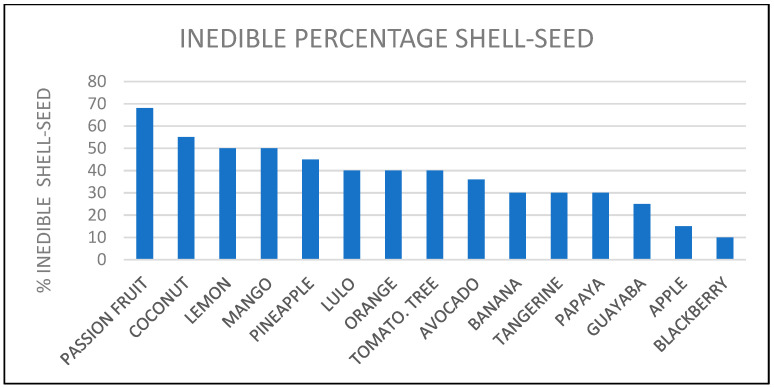
Inedible percentage shell–seed of fruits in Colombia. Reprinted with permission from ref [21]. 2018 ICBF.

**Figure 3 molecules-27-02124-f003:**
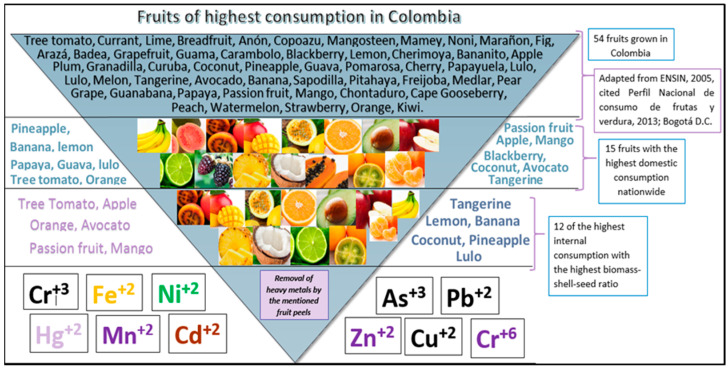
The pyramid of the most consumed fruits in Colombia [20]. Own elaboration.

**Figure 4 molecules-27-02124-f004:**
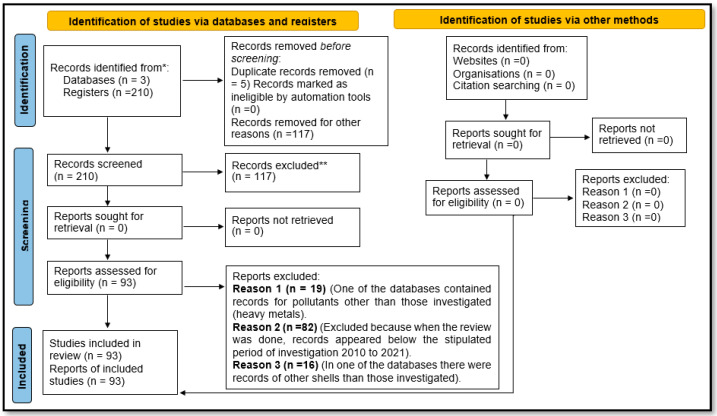
PRISMA format of selected articles in the Scopus, Web of Science, Science Direct databases from 2010 to 2021.

**Figure 5 molecules-27-02124-f005:**
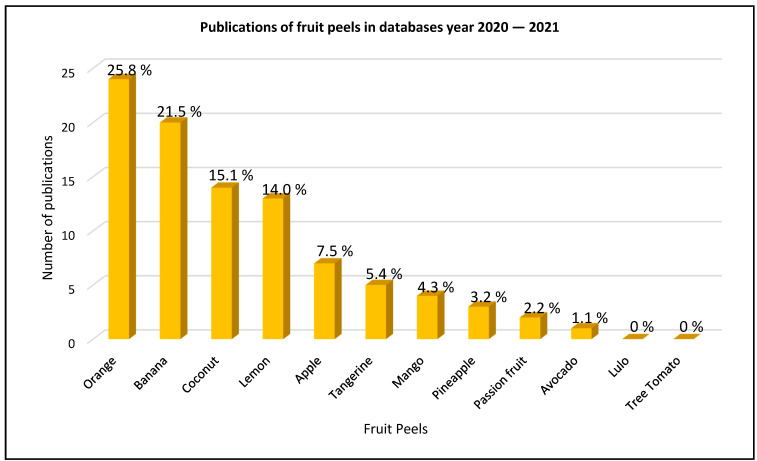
Compilation of selected fruit peels in Scopus, Web of Science, and Science Direct databases from 2010–2021.

**Figure 6 molecules-27-02124-f006:**
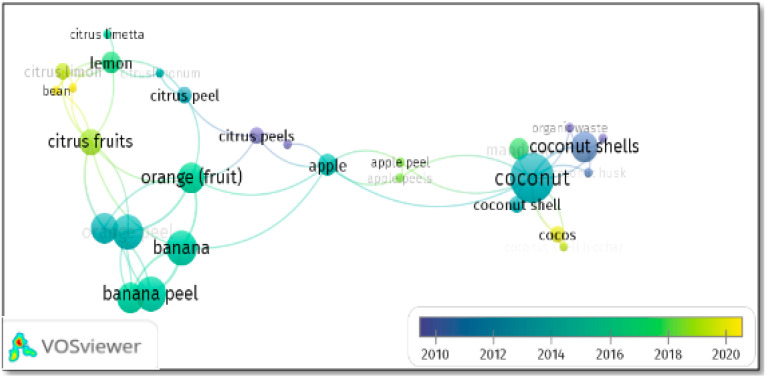
Co-occurrence of fruit peels with respect to publications of heavy metals removal in wastewater. Taken from Scopus/VOSviewer 1.6.17 metadata (https://www.vosviewer.com/getting-started, accessed on 18 January 2021).

**Figure 7 molecules-27-02124-f007:**
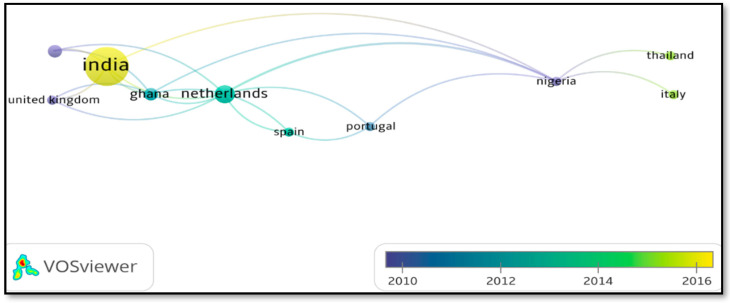
Co-occurrence of countries with respect to heavy metals publications. Taken from Scopus/VOSviewer 1.6.17 metadata.

**Figure 8 molecules-27-02124-f008:**
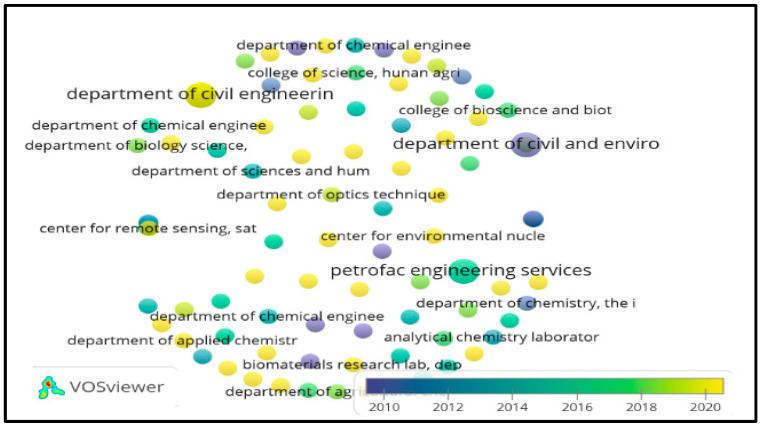
Co-occurrence of authors presenting publications on heavy metals removal in wastewater. Taken from Scopus/VOSviewer 1.6.17 metadata.

**Figure 9 molecules-27-02124-f009:**
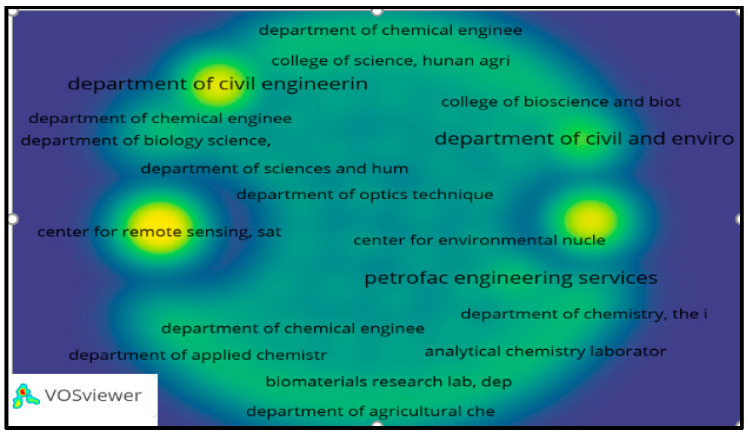
Heat map of organizations presenting publications on heavy metals removal in wastewater. Taken from Scopus/VOSviewer 1.6.17 metadata.

**Figure 10 molecules-27-02124-f010:**
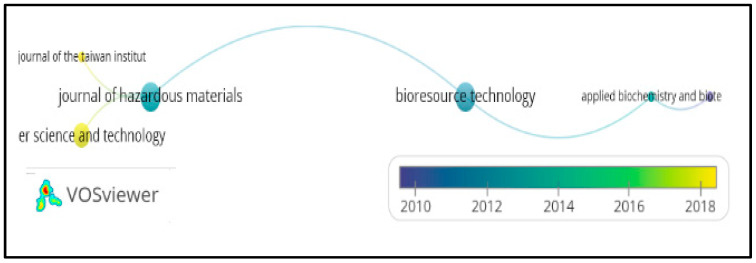
Co-occurrence of journals presenting publications on heavy metals removal in wastewater. Taken from Scopus/VOSviewer 1.6.17 metadata.

**Figure 11 molecules-27-02124-f011:**
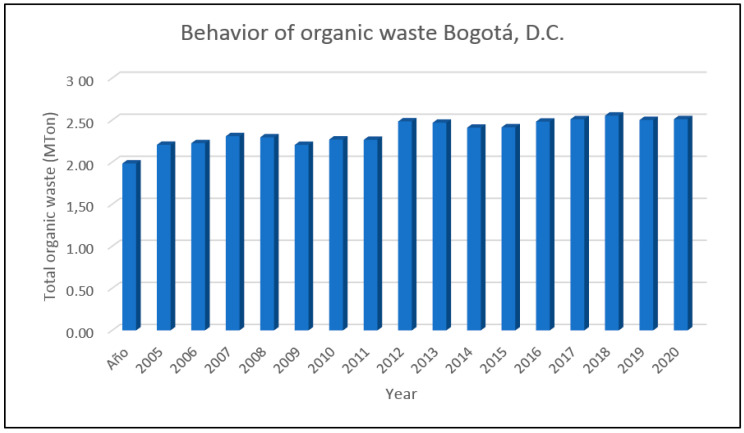
Organic solid waste behavior in Bogotá, D.C. Reprinted with permission from ref [117]. 2016 UAESP.

**Figure 12 molecules-27-02124-f012:**
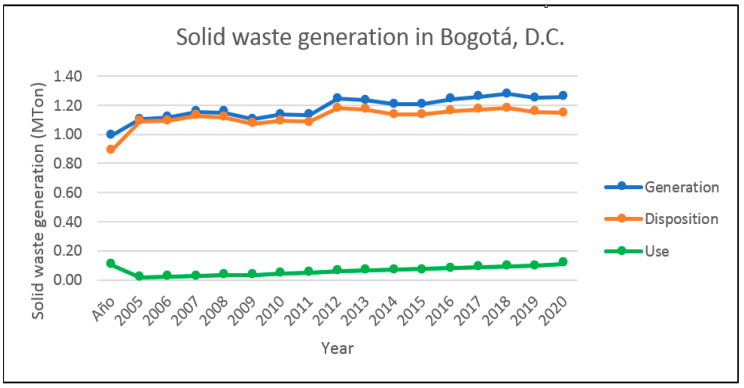
Solid waste production in Bogotá, D.C. Reprinted with permission from ref [117]. 2016 UAESP.

**Figure 13 molecules-27-02124-f013:**
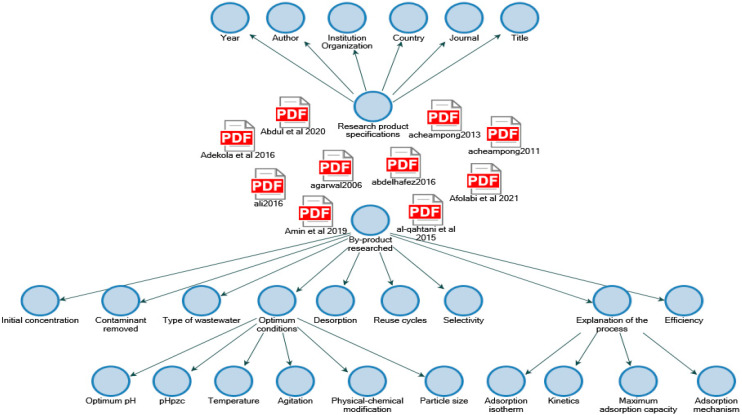
Content analysis map. Source: Elaborated with NVivo 12 Plus.

**Figure 14 molecules-27-02124-f014:**
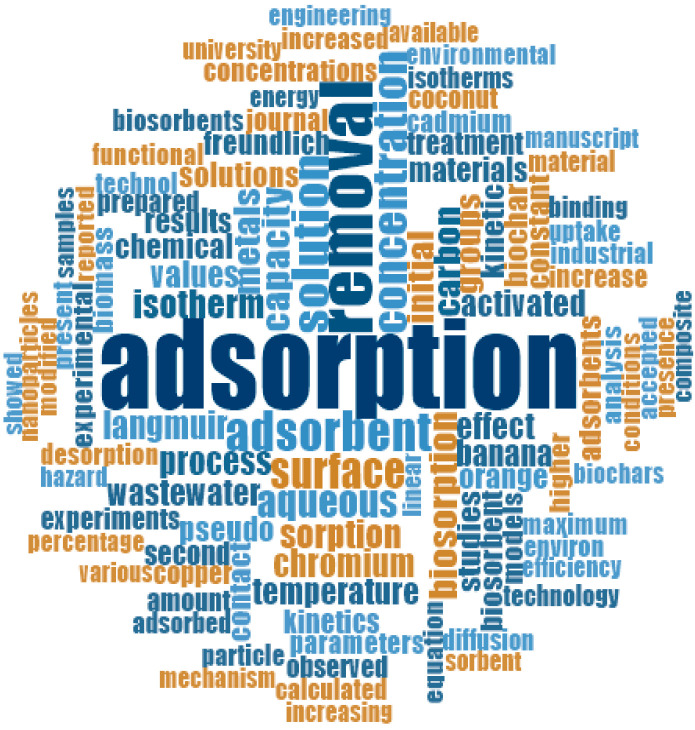
Word cloud. Source: Elaborated with NVivo 12 Plus.

**Figure 15 molecules-27-02124-f015:**
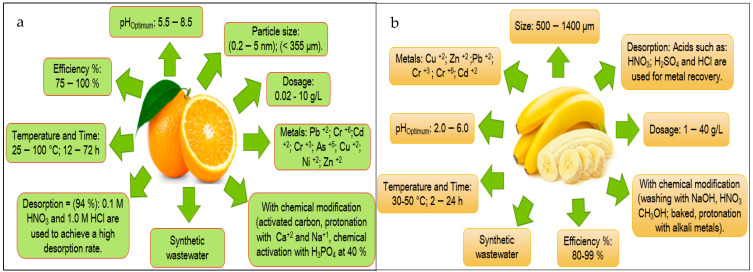
Fruit peels. (**a**) optimal orange peel conditions; (**b**) optimal banana peel conditions.

**Figure 16 molecules-27-02124-f016:**
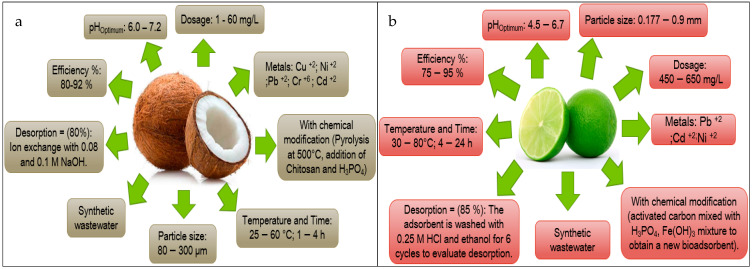
Fruit peels. (**a**) Optimal conditions of lemon peel; (**b**) optimal conditions of coconut peel.

**Figure 17 molecules-27-02124-f017:**
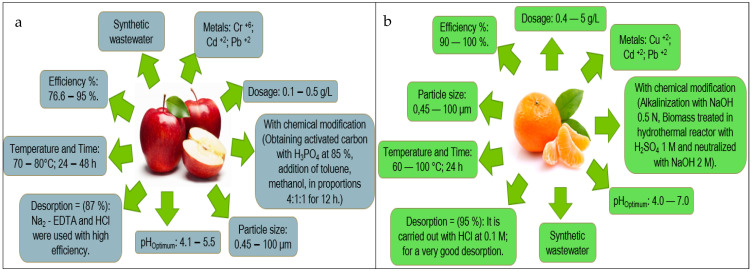
Fruit peels. (**a**) Optimal conditions of Apple peel; (**b**) optimal conditions of tangerine peel.

**Figure 18 molecules-27-02124-f018:**
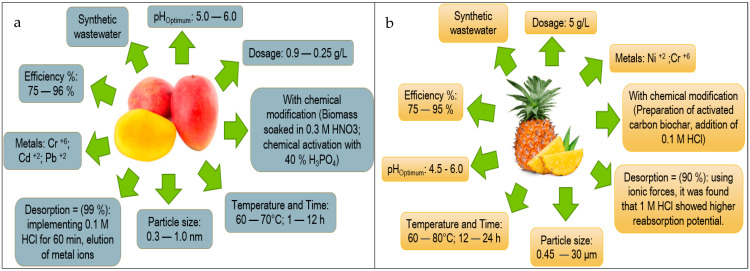
Fruit peels. (**a**) Optimal conditions of mango peel; (**b**) optimal conditions of pineapple peel.

**Figure 19 molecules-27-02124-f019:**
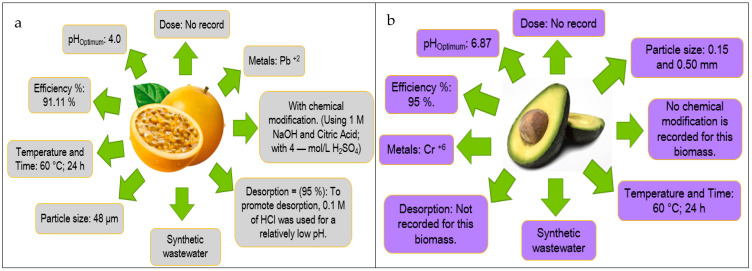
Fruit peels. (**a**) Optimal conditions of passion fruit peel; (**b**) optimal conditions of avocado peel.

**Figure 20 molecules-27-02124-f020:**
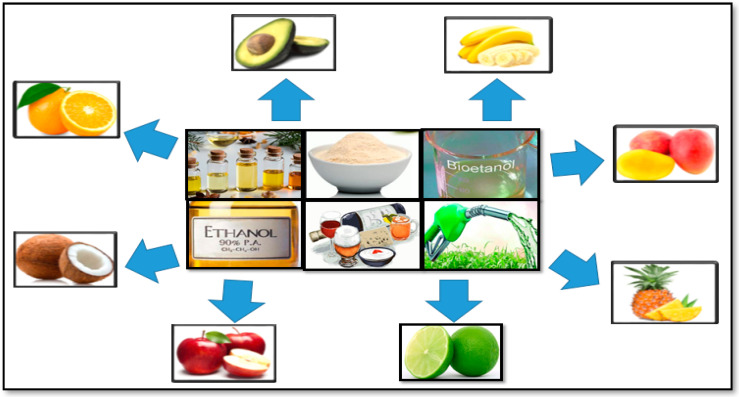
Biorefinery of fruit peels [118,129]. 2022, own elaboration.

**Figure 21 molecules-27-02124-f021:**
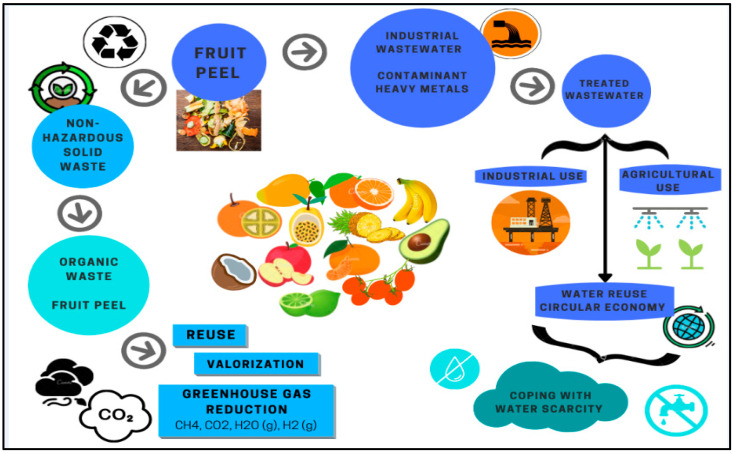
Evaluation of fruit peels in the removal of heavy metals in industrial wastewater, own elaboration.

**Figure 22 molecules-27-02124-f022:**
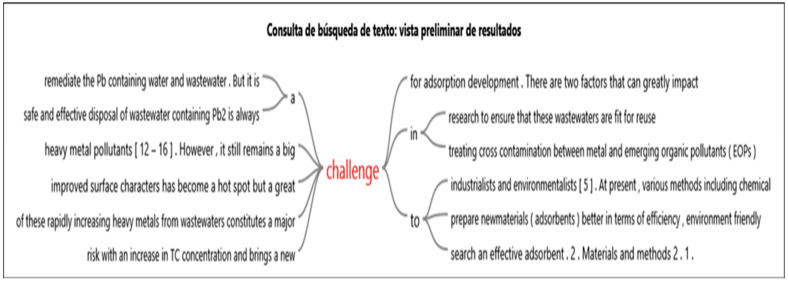
Word tree of challenges from the use of fruit peels. Source NVivo 12 plus.

**Table 1 molecules-27-02124-t001:** Consolidation of the 12 fruit peels in Scopus, Web of Science, and Science Direct databases from 2010 to 2021.

Fruit Peel	Scopus (Number of Articles)	References Scopus	Science Direct (Number of Articles)	References Science Direct	Web of Science (Number of Articles)	References Web of Science
Orange	9	[22,23,24,25,26,27,28,29,30]	2	[31,32]	13	[9,33,34,35,36,37,38,39,40,41,42,43,44]
Banana	8	[45,46,47,48,49,50,51,52]	7	[53,54,55,56,57,58,59]	5	[9,50,55,60,61]
Coconut	6	[62,63,64,65,66,67]	2	[68,69]	6	[66,70,71,72,73,74]
Lemon	5	[29,30,75,76,77]	5	[78,79,80,81,82]	3	[82,83,84]
Apple	3	[29,85,86]	3	[85,87,88]	1	[89]
Tangerine	1	[90]	1	[57]	3	[91,92,93]
Mango	0	-	3	[94,95,96]	1	[97]
Pineapple	0	-	3	[98,99,100]	0	-
Passion fruit	0	-	1	[101]	1	[102]
Avocado	0	-	0	-	1	[103]
Lulo	0	-	0	-	0	-
Tree Tomato	0	-	0	-	0	-

**Table 2 molecules-27-02124-t002:** Generalities of the selected fruits in Colombia [19].

General Information on Selected Fruits in Colombia			
Species	Origin/Year	Type of Fruit	References
Lemon	China/1941	Major perennials	[104]
Banana	Asia/NE	Transient fruits	[105]
Mango *	India/NE	Major perennials	[106]
Tree tomato	Peru/NE	Transient Fruits	[107]
Pineapple	Southeast Brazil and Paraguay/NE	Transient fruits	[108]
Passion fruit	Brazil/NE	Transient fruits	[109]
Orange *	China/NE	Major perennials	[110]
Coconut	Malaysia/NE	Major perennials	[111]
Avocado *	Mexico and Guatemala/NE	Major perennials	[112]
Apple *	Central Asia/NE	-	[113]
Lulo	Colombia/NE	Transient fruits	[114]
Tangerine *	Asia/NE	Major perennials	[115]

* Fruits of greater daily consumption.

**Table 3 molecules-27-02124-t003:** Organic characterization of orange peel. Reprinted with permission from ref [123]. 2014 Tejada.

Parameter	Value (%)	Analytical Method of Quantification	Reference
Carbon	42.04	AOAC 949.14	[123]
Hydrogen	5.44	AOAC 949.14	
Nitrogen	0.70	AOA 984.13 Kjeldahl	
Pectin	10.98	Acid digestion and Thermogravimetry	
Lignin	6.51	Photocolorimetry	
Cellulose	13.08	digestion and Thermogravimetry	
Hemicellulose	6.47	digestion and Thermogravimetry	

**Table 4 molecules-27-02124-t004:** Inorganic composition of orange peel by X-ray fluorescence analysis. Reprinted with permission from ref [124]. 2013 Mafra.

Characteristics	Values	Reference
CaO	1.42	[124]
K_2_O	0.18	
SO_3_	0.14	
MgO	0.12	
Fe_2_O_3_	0.11	
SiO_2_	0.08	
P_2_O_5_	0.05	
BaO	0.02	
SrO	0.01	
Al_2_O_3_	0.01	
NiO	0.01	
WO_3_	Not detected	
ZnO	Not detected	
Mn	Not detected	

**Table 5 molecules-27-02124-t005:** Characterization of functional groups in orange peel, infrared spectrum.

Functional Group	Displacement cm^−1^	References
Hydroxyl groups (OH) ranging from 3340 to 3600 cm^−1^.	3441	[125,126,127]
(C-H) methyl, methylene, and methoxy groups.	2923.78	
Carbonyl (C=O), indicating the vibration of the carboxyl groups of pectin.	1748.15	
Stretching of (C=C), as a consequence of the presence of aromatic rings.	1636.17	
Presence of (C-H), aliphatic and aromatic, groups in a deformation plane, methyl, methylene, and methoxy groups.	1444.43	
The range corresponds to the (C-O) group of alcohols and carboxylic acids.	1333.24–1022	

**Table 6 molecules-27-02124-t006:** Industrial activities that generate heavy metals [10,128]. 2015, Caviedes, D.I.; Muñoz, R.A.; Perdomo, A.; Rodriguez, D.; Sandoval, I.J.

Industry	Metals Generated by the Activity	Contamination By-Products
Ferrous metal mining	Cd, Cu, Ni, Cr, Co, Zn	Acid mine drainage, tailings, tailings dumps, ferrous metals and steel mills, chemical industry.
Ore extraction	As, Cd, Cu, Ni, Pb, Zn	Presence in ores as well as in by-products.
Smelting	As, Cd, Pb, Ti	Ore processing to obtain metals.
Metallurgy	Cr, Cu, Mn, Pb, Sb, Zn	Thermal processing of metals.
Alloys and steels	Pb, Mo, Ni, Cu, Cd, As, Te, U, Zn	Metal fabrication, disposal and recycling Tailings.
Waste management	Zn, Cu, Cd, Pb, Ni, Cr, Hg, Mn	Waste incineration or in leachates.
Metal corrosion	Fe, Cr, Pb, Ni, Co, Zn	Instability of metals exposed to the environment.
Electroplating	Cr, Ni, Zn, Cu	Liquid effluents from coating processes.
Paints and pigments	Pb, Cr, As, Ti, Ba, Zn	Aqueous waste from the manufacture and deterioration of old paint.
Batteries	Pb, Sb, Zn, Cd, Ni, Hg	Waste pile fluid, soil and groundwater contamination.
Electronics	Pb, Cd, Hg, Pt, Au, Cr, As, Ni, Mn	Aqueous and solid metal waste from the manufacturing and recycling process.
Agriculture and livestock	Cd, Cr, Mo, Pb, U, V, Zn, As, Mn, Cu	Contamination of runoff, surface and ground water, plant bioaccumulation.

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
