# Peer review of "Fruit Peels as a Sustainable Waste for the Biosorption of Heavy Metals in Wastewater: A Review"

_molecules, 2022, doi:10.3390/molecules27072124_

Round 1
Reviewer 1 Report
Peer-reviewed manuscript is devoted to the topical problem of using fruit peels as a sustainable waste in the process of removing heavy metals from industrial wastewater. The manuscript is well written and structured. There are several questions and comments:
1) Are the authors aware of industrial use fruit peel cases for the removal of heavy metals from wastewater? It would be advisable to provide examples.
2) Taking into account the large dosage of adsorbents (Figures 15-19), its applicability for the treatment of large volumes of wastewater (mining, metallurgy, etc.) should be discussed.
3) The questions of heavy metals desorption efficiency and waste bioadsorbents utilization have not been fully discussed.
4) It would be advisable to discuss the advantages of the non-traditional sorbents proposed.

Author Response
Peer-reviewed manuscript is devoted to the topical problem of using fruit peels as a sustainable waste in the process of removing heavy metals from industrial wastewater. The manuscript is well written and structured. There are several questions and comments:
1) Are the authors aware of industrial use fruit peel cases for the removal of heavy metals from wastewater? It would be advisable to provide examples.
|
In relation to this question, it is important to clarify two important points: a) Fruit peels have been used as adsorbents for the removal of various metals using industrial wastewater as an aqueous matrix, among the most used are reported in the electroplating industry for the removal of copper (N. Feng, X. Guo, S. Liang, Adsorption study of copper (II) by chemically modified orange peel, J. Hazard. Mater. 164 (2009) 1286–1292), remoción de Molibdeno (W. Shan, D. Fang, Z. Zhao, Y. Shuang, L. Ning, Z. Xing, Y. Xiong, Application of orange peel for adsorption separation of molybdenum(VI) from re-containing industrial effluent, Biomass Bioenergy 37 (2012) 289–297), remoción de Cr (VI) (R.A.K. Rao, F. Rehman, M. Kashifuddin, Removal of Cr(VI) from electroplating wastewater using fruit peel of Leechi (Litchi chinensis), Desalin. Water Treat.49 (2012) 136–146) (Remoción Cu, Cd, Pb) (C.S. Inagaki, T.d.O. Caretta, R.V.d.S. Alfaya, A.A.d.S. Alfaya, Mexerica mandarin (Citrus nobilis) peel as a new biosorbent to remove Cu(II), Cd(II), and Pb(II) from industrial effluent, Desalin. Water Treat. 51 (2013) 5537–5546. Finally, some publications have also been found in the treatment of the tannery industry (Remoción Cr (VI)) Monroy F, Echavarría M, Gómez D. Design and validation of a system for hexavalent chromium adsorption in tannery effluents using orange peel and wheat bran. ,Tecnología y ciencias del agua, ISSN 2007-2422, 12(3), 1-31. DOI: 10.24850/j-tyca-2021-03-01 It is relevant to note that these waters have only been used in pilot-type experimental designs using the optimal adsorption conditions and have not yet been implemented on an industrial scale, thus becoming a challenge at the technological level, some authors argue that a significant financial and technological effort is required to scale it up to a truly large scale. (Amit Bhatnagar, Mika Sillanpaa, Anna Witek-Krowiak). On the other hand, it is important to highlight those peels have been used for the removal of other pollutants (colorants and organic matter) present in domestic and industrial wastewater. b) With respect to the article, it was specified that the most frequently reported peels in the databases consulted have been orange, banana and coconut peels, all of which have only been used with synthetic wastewater. For this reason, it becomes a future opportunity for design on an industrial scale. In other words, with respect to this comment, the authors incorporated the information in section 5 of the article on techno-economic aspects and future research. 2) Taking into account the large dosage of adsorbents (Figures 15-19), its applicability for the treatment of large volumes of wastewater (mining, metallurgy, etc.) should be discussed. With respect to this question, it is related to the previous paragraph; since only some of these fruit peels have been implemented on a pilot scale with industrial wastewater (electroplating, battery industry and tanneries). Therefore, there is still a lack of research on industrial wastewater from mining, metallurgy, glass, chemicals, wood protection, plastics, paper, rubber, alloys and electronics industries. The research would consist of: First simulate the optimal adsorption conditions that are compiled in Figures 15-19; given that, these were used with synthetic and not real wastewater. Therefore, it is necessary to make the respective adjustments to determine the minimum adsorbent dose, contact time, adsorbent particle size, agitation, temperature, metals to be removed and the efficiency of the process. Secondly, when these conditions are already specified, whether to take it to industrial scale. For this purpose, it is important to mention R.A.K. Rao, F. Rehman, M. Kashifuddin, Removal of Cr(VI) from electroplating wastewater using fruit peel of Leechi (Litchi chinensis), Desalin. Water Treat.49 (2012) 136–146; since, they conclude that 6 g/L of the biaodsorbent with the conditions established in the article is sufficient to remove 100% Cr(VI) in industrial electroplating wastewater. 3) The questions of heavy metals desorption efficiency and waste bioadsorbents utilization have not been fully discussed. This suggestion was incorporated by the authors in section 5 of the article "techno-economic aspects and future research". 4) It would be advisable to discuss the advantages of the non-traditional sorbents proposed. This suggestion was incorporated by the authors in section 5 of the article "techno-economic aspects and future research".
|

Reviewer 2 Report
The article gives a documentary analysis on fruit Peels as biosorption materials and its adsorption of heavy metals in wastewater. The review is clear and comprehensive. The conclusions based on documentary and content analysis are helpful to the research on bisorption of heavy metals. This could be accepted with a minor version and the following comments could be considered.
1.The content from line 135 to line 174 could be shortened.
2.Line199 to line200 should be deleted.
3.Table 2 could be deleted and some data could be showed in Figure 5.
The above comments are drawn only based the paper contents. The paper was submitted to the Special Issue of Residues of Organic Pollutants in Environmental Samples in section of Analytical Chemistry. The topic of the paper don’t belong to the issue. This could be considered by the editor.
Author Response
- The content of line 135 to line 174 could be shortened.
We appreciate the comment, but the authors specify that this information is relevant to contextualize the reader, on the selection of the 12 fruit peels of greater national consumption in Colombia which were found in systematic review analysis by using databases to highlight that fruit peels become a research gap at the level of heavy metals removal.
- Lines 199 to 200 should be deleted.
The change suggested by the referee was included.
- Table 2 could be removed and some data could be shown in Figure 5.
The change suggested by the referee was included.

Reviewer 3 Report
The manuscript entitled “Fruit Peels as a Sustainable Waste for the Biosorption of Heavy Metals in Wastewater: A Review” aims to give us an overview of the scientific articles in different databases published from 2010 to 2021 on the use of fruit peels as a sustainable waste in the removal of heavy metals from industrial wastewater. This topic is very important and interesting. Still, the approach in this manuscript is not scientific enough. The manuscript focuses insufficiently on the connection between the structure and performance of the reviewed materials derived from the peels. The discussion on that part has to be improved. The manuscript is also too long and has many unnecessary Figures, like no. 6, 7, 8, 9, 10, 13, 14, and 20.
Author Response
- The manuscript insufficiently focuses on the connection between structure and performance of the reviewed peel-derived materials. We need to improve the discussion on that part.
With respect to this comment, the chemical characterization of both organic and inorganic compounds present in the orange peel was expanded (see Tables 3 and 4). At the same time, the displacements of the functional groups present in the orange peel were introduced from an Infrared report (see Table 5), with the objective of specifying which of these are responsible for the interaction of the lignocellulosic material with heavy metals.
2. The manuscript is also too long and has many unnecessary Figures, of course. 6, 7, 8, 9, 10, 13, 14 and 20.
Regarding this comment, the authors specify that the figures cited by the reviewer are relevant to the structure of the article, given that one objective of the article is to carry out a documentary analysis to investigate the countries, authors, journals of publication and optimal conditions of adsorption by the selected fruit peels.

Round 2
Reviewer 3 Report
Regarding the answer to my second comment, the authors cannot convince me that, for example, Figure 14 has a scientific value and contains important information. The manuscript is too long, at least the number of Figures should be reduced. They can be transferred to Supplementary material.
Author Response
The authors thank you again for the comment and in this regard, the following is argued:
Figure 14 refers to the word cloud that relates the triangulation of the information found, using the Nvivo 12 plus software in which the co-occurrence is observed, to determine the terms that were most employed by the authors of the articles found; to obtain emergent or abductive categories to analyze aspects such as the optimal conditions for adsorption of heavy metals in industrial wastewater. To this end, the importance of the aforementioned is specified in the article in lines 323 and 324.
On the other hand, with respect to the fact that the manuscript is too long, the authors express that the information compiled in it is very relevant since first a documentary search was made of those fruit peels that are currently being generated in Colombia and which have been used as bioadsorbent material to remove heavy metals in industrial wastewater, thus obtaining information that was necessary to record them in figures and tables that are of interest to the readers. Therefore, we still think that all the information gathered in the manuscript is relevant.
Finally, the figures summarize in a more understandable way for the reader the optimal adsorption and desorption variables for each agricultural residue. It also allows the reader to observe relationships between countries, authors and publication journals that may be of help and/or interest to research groups worldwide.